# Does Single or Combined Caffeine and Taurine Supplementation Improve Athletic and Cognitive Performance without Affecting Fatigue Level in Elite Boxers? A Double-Blind, Placebo-Controlled Study

**DOI:** 10.3390/nu14204399

**Published:** 2022-10-20

**Authors:** Murat Ozan, Yusuf Buzdagli, Cemre Didem Eyipinar, Nurcan Kılıç Baygutalp, Neslihan Yüce, Furkan Oget, Emirhan Kan, Fatih Baygutalp

**Affiliations:** 1Department of Physical Education and Sports, Kazım Karabekir Faculty of Education, Atatürk University, 25500 Erzurum, Turkey; 2Department of Coaching Education, Faculty of Sport Sciences, Erzurum Technical University, 25500 Erzurum, Turkey; 3Department of Physical Education and Sport, Faculty of Sport Sciences, Gaziantep University, 27310 Gaziantep, Turkey; 4Department of Biochemistry, Faculty of Pharmacy, Ataturk University, 25500 Erzurum, Turkey; 5Department of Medical Biochemistry, Faculty of Medicine, Ataturk University, 25500 Erzurum, Turkey; 6Department of Physical Education and Sports, Faculty of Sport Sciences, Erzurum Technical University, 25500 Erzurum, Turkey

**Keywords:** caffeine, taurine, cognitive, balance, agility

## Abstract

In previous studies, the effect of single or combined intake of caffeine (CAF) and taurine (TAU) on exercise performance was investigated. However, the potential synergistic effect on physical and cognitive performance after fatigue induced by anaerobic exercise is unknown. The effects of single and combination CAF and TAU supplementation on the Wingate test in elite male boxers and to evaluate balance, agility and cognitive performance after fatigue are being investigated for the first time in this study. Twenty elite male boxers 22.14 ± 1.42 years old were divided into four groups in this double-blind, randomized crossover study: CAF (6 mg/kg of caffeine), TAU (3 g single dose of taurine), CAF*TAU (co-ingestion of 3 g single dose of taurine and 6 mg/kg of caffeine) and PLA (300 mg maltodextrin). The findings are as follows: co-ingestion of CAF*TAU, improved peak (W/kg), average (W), minimum (W) power, time to reach (s), and RPE performances compared to the PLA group significantly (*p* < 0.05). Similarly, it was determined that a single dose of TAU, created a significant difference (*p* < 0.05) in peak power (W/kg), and average and minimum power (W) values compared to the CAF group. According to the balance and agility tests performed after the Wingate test, co-ingestion of CAF*TAU revealed a significant difference (*p* < 0.05) compared to the PLA group. In terms of cognitive performance, co-ingestion of CAF*TAU significantly improved the neutral reaction time (ms) compared to the TAU, CAF and PLA groups. As a result, elite male boxers performed better in terms of agility, balance and cognitive function when they consumed a combination of 6 mg/kg CAF and 3 g TAU. It has been determined that the combined use of these supplements is more effective than their single use.

## 1. Introduction

For many years, elite athletes have aimed to find the only thing that can give them “superiority” in the competition. In this context, supplements are becoming indispensable for elite athletes who want to gain superiority and competitive advantage over their competitors [1,2,3].

Caffeine (CAF) is a naturally derived stimulant typically found in many dietary supplements such as guarana, nut, or beverages such as cola, coffee, and tea. Many studies have also shown that CAF is an effective ergogenic aid for aerobic and anaerobic exercise [4]. Studies show that caffeine intake (e.g., 3–9 mg/kg 30–90 min before exercise) can reduce carbohydrate use, thereby improving endurance exercise capacity [5,6,7]. In addition to the apparent positive effects on endurance performance, caffeine has also improved repeated sprint performance by benefiting athletes engaged in anaerobic branches of sport [7,8,9]. Findings of studies examining the ability of caffeine to increase maximum strength and repetition against fatigue are considered appropriate to increase the evidence for various sports, including combat sports [9,10]. Research on caffeine has evaluated its ability to affect athletic performance, but research findings are unclear. In brief, even though scientific evidence shows that caffeine is an effective supplement, it cannot be ignored that it should be expanded on a consistent scientific basis.

Taurine (TAU) is an abundant human skeletal muscle amino acid derived from cysteine metabolism that plays a role in various physiological functions [11,12]. Taurine benefits various metabolic and physiological processes [13,14]. As a result, taurine has been considered as a potential ergogenic supplement to enhance athletic performance [15,16]. When exercise is completed, plasma taurine concentrations reach their highest [17]. The increased release of taurine is most likely caused by the involvement of taurine in the control of Ca^2+^ use during contraction in skeletal and cardiac muscle, in addition to the enhanced responsiveness of force-generating myofilaments to calcium [18,19].

As a result, taurine release controls the concentration of channels and transporters, which in turn influences calcium uptake, exercise performance, and fat metabolism. In recent years, the effect of taurine intake on various exercise performances has been studied. Based on previous studies, the ergogenic effects of taurine administration remain controversial. While the studies on taurine are evaluated in terms of its ability to affect sport performance, some studies show that it has no effect [20,21], and some studies report its positive effect [22,23,24,25]. As more research on taurine supplementation continues to be published, there remains confusion about taurine’s potential to enhance athletic performance. Recent results suggest that co-ingestion of caffeine and taurine may improve peak and mean power output during repeated Wingate performances compared to placebo [22]. However, the doses used are much higher than the doses contained in the available energy drinks on the market. In addition, the co-ingestion of caffeine with taurine has a potential negative performance of taurine generation after generation due to in vivo interactions. Beverage effects on exercise performance have been the subject of several research studies, with a variety of outcomes.

Combining caffeine and taurine at the levels found in energy drinks increased alertness, aerobic and anaerobic performance, reaction time, simulated football performance, and upper body strength [22,26,27,28], and others, during repeated Wingate tests, failed to improve time to exhaustion, peak power output or repeated sprint running ability [10,29]. The majority of these trials, though, did not use separated dosages of the ergogenic compounds (caffeine and taurine) and thus did not consider other ingredients’ contributions.

Successful performance in modern sports, especially boxing, is determined by many factors. It addresses various performance areas for optimum performance at an elite level [30,31]. In boxing, which is an Olympic sport, it is of great importance to be able to resist fatigue, tolerate this fatigue, and protect balance [32,33], agility [34,35] and cognitive performance [36,37], which play a vital role in boxing. This study evaluated anaerobic power, balance, agility and cognitive performance characteristics as significant predictors of boxing performance. Furthermore, it is unclear whether caffeine, taurine, co-ingestion of caffeine and taurine by elite athletes participating in anaerobic sports (e.g., boxing) can be considered ergogenic and alters the balance, agility and cognitive performance after exercise. Hence, the aims of this study were (a) to evaluate the ergogenic effect of caffeine, taurine and the co-ingestion of caffeine and taurine on the Wingate test in elite boxers and (b) to determine the effects and possible changes in these parameters on balance, agility and cognitive performance after exercise.

## 2. Materials and Methods

### 2.1. Participants

Twenty elite male boxer athletes (age: 22.14 ± 1.42 years, sports age: 11.12 ± 1.12 years, height: 178 ± 6.45 cm, body mass: 75.45 ± 8.95 kg) participated in this study (Table 1). The inclusion criteria were as follows: (a) being over 18 years old and (b) having at least 10 years of experience in boxing and national and international degrees. The exclusion criteria were as follows: (a) consumption of any substance such as nutritional supplements or steroids in the last three months that may affect hormone levels or sports performance, (b) consumption stimulants, narcotics and/or psychoactive substances throughout the test or supplementing phase, and (c) a determination of any orthopedic, neurological, cardiovascular, pulmonary, metabolic condition that might impair performance on various tests. Prior to completing the informed permission form, participants were provided with information regarding the research protocol, timetable and types of exercises and assessments they would be required to complete. All protocols and procedures were carried out according to the Declaration of Helsinki and were approved by the Erzurum Atatürk University Ethics Committee (E-70400699-050.02.04-2100151951, May 2022 2022/5).

### 2.2. Study Design

Five testing sessions composed this double-blind, randomized, crossover study: a familiarization session, anthropometric measures, four experimental trials with separate CAF or TAU, a combination of the two agents (CAF*TAU), and a placebo (PLA). To avoid bias from circadian rhythm interference associated with supplement intake, participants participated four times in a 72 h window in the same time frame (±0.5 h) [38]. In all four sessions, participants were randomly divided into placebo, caffeine, taurine and caffeine*taurine supplementation. Wingate (WT), lactate, balance, agility and cognitive performance measures were administered in each experimental session to assess anaerobic, athletic and cognitive improvement. The pretest was performed in a separate session without supplementation (Figure 1).

### 2.3. Supplementation Protocol

The doses were calculated by using an analog scale sensitive to 1 mg, and all supplements were supplied in powder form. The compounds were swallowed with tap water after being enclosed in indistinguishable gelatin capsules. The capsules contained either 300 mg of maltodextrin as a PLA or 6 mg/kg of CAF, 3 g of TAU, or 6 mg/kg of CAF and 3 g of TAU.

The same company (My Protein, Manchester, UK) provided all of the supplements. Supplements were taken 60 min before the start of the Wingate test. Based on suggestions from the International Society of Sports Nutrition’s position on CAF [39], the CAF dose was chosen. The TAU dosage was chosen due to its being characteristic of the dosage found in energy drinks already on the market [40]. The 60 min window was selected to allow for direct comparison with earlier research and to generally reflect the peak plasma availability of TAU [41] and CAF [39]. A diet list was given to the participants to ensure a similar nutrient intake (60% carbohydrate, 30% lipid and 10% protein) until 72 h before the start of the study. In addition, the intake of caffeine and stimulant drugs was also limited to 48 h before the experimental session to avoid potential interference with the study results for supplement elimination. For this reason, a list was prepared for participants to avoid caffeine-rich foods such as coffee, tea, mate, energy drinks, cola drinks, chocolate drinks and chocolate.

### 2.4. Wingate Anaerobic Test

The Wingate was performed on a specialized cycle ergometer (Monark 894E, Peak Bike, Vansbro, Sweden). In the familiarization session, the seat and handle positions were modified for each participant (with a knee angle of approximately 170–175°), and they were repeated in the remaining test sessions. The participants’ legs were kept securely in position and touched with the pedals thanks to the usage of toe clamps. After a five-minute warm-up at 60 W that also included five-second sprints without resistance at the second and third minutes, participants had to pedal from a stationary start for 30 s while also exerting their maximum effort against a fixed load of 7.5% of their body mass, as recommended [42]. Additionally, participants were instructed to crank with minimal body rotation and without pedal acceleration (initial speed was zero) [43]. Solid verbal encouragement and standard practice were used to ensure that they performed the tests to their maximum performance.

### 2.5. Rating of Perceived Exertion (RPE)

The Borg Scale (6–20), which is known in the literature, is a valuable indicator for subjectively monitoring the exercise tolerance of individuals. The Borg scale of fatigue felt by the participant during the exercise was nothing (6), very, very light (7–8), very light (9–10), light (11–12), somewhat difficult (13–14), difficult (15–16), very difficult (17–18), very, very difficult (19), and exhausted (20). It is primarily used to monitor the person’s progress to maximal effort during the exercise test [44]. The Borg scale was applied immediately after each Wingate test measurement.

### 2.6. Blood Lactate Levels Measured

The Lactate Scout 4 (Leipzig, Germany) analyzer was used to collect blood samples (5 µL) from the tip of the left hand’s index finger according to manufacturer’s instructions.

### 2.7. Balance Test

Static and dynamic balance measurements were made with the “SPORT KAT 4000 Balance Measurement” device. This device can make both static and dynamic balance measurements. Before each measurement, the platform was calibrated. The best result was selected in the studies following an evaluation of the consistency of the measurements following the manufacturer’s recommendations. Before the measurements were performed, the participants were verbally explained the information about the balance device and the test during the familiarization process. Afterwards, the athletes were allowed to try static and dynamic balance tests to know the device and be more efficient in the test. Dominant, nondominant, and both-leg measurements were taken in static and dynamic balance tests.

### 2.8. Agility Test (Illinois)

It is a test track consisting of four cones lined up on a straight line at intervals of 5 m in width, 10 m in length, and 3.3 m in the middle. The test consists of a 40 m straight and 20 m slalom run between cones with 180° turns every 10 m After the test track was prepared, a two-gate photocell electronic chronometer system (Tumer Elektronik Ltd., Ankara, Turkey) was installed at the start and end points for measurement with an accuracy of 0.01 s. The subjects exited the starting line of the test track while they were ready for the running position. The time to finish the track was recorded in seconds, and the test was performed once [45].

### 2.9. Stroop Test

The Stroop test is a neuropsychological test that reflects frontal region activity. It was discovered that it takes more time to say the names of objects or colors than to read the words associated with them, and it has been shown that the event is a “color–word interference effect” [46]. The Stroop task consists of three parts: neutral, congruent, and incongruent. Participants were asked to press the “←” or “→” direction button with their right index and ring fingers to respond. The reaction time and error rate were measured. The Stroop task consists of 4 blocks, with 30 neutral, 30 congruent and 30 incongruent. At the start and end of the task, the baseline was taken for 45 s, and the stimulus remained on the screen until a response was given or for 2000 ms. Stimuli were given at 1000 ms intervals. For an acceptable response, responses are given within 200 and 2000 ms after the presentation of the stimulus were considered correct. Responses that were not within the time range (i.e., 200–2000 ms) and responses that were given when the participant pressed the wrong color button were considered incorrect. All words are written in Turkish characters. The Stroop task was designed in Psychtoolbox for MATLAB 2018.

### 2.10. Familiarization

We carefully selected each participant within the scope of the project. Due to their elite athlete status, participants were already familiar with the protocol for the Wingate test and the tests of balance, agility and cognition. However, all participants were informed about the application/trial process and the exercise to be performed to avoid any problems. Then, 3 days before the start of the exercise session, all participants were recruited to administer the Wingate, balance, agility and cognitive testing protocols. Thus, it went through the habituation/trial process. Within the project’s scope, a predetermined visualized experiment flow chart was created for this study. All experimental procedures are summarized.

### 2.11. Statistical Analyses

Data are presented as the mean ± standard deviation (SD). The Shapiro–Wilk test was used to determine whether all the variables obtained showed a normal distribution. An analysis of variance over time (ANOVA-RM) was applied for repeated measurements. Greenhouse–Geiser corrections for nonspherical distributions were evaluated using the Mauchly test, and Bonferroni corrections were applied for post hoc comparisons. Partial eta squared was calculated for ANOVA-RM, where <0.25, 0.26–0.63 and >0.63 were considered small, medium and large effect sizes, respectively [47]. Tests were conducted using IBM SPSS Statistics for 25.0 (IBM Corp., Armonk, NY, USA). The significance level was determined as *p* ≤ 0.05. Biorender (https://biorender.com/ accessed on 1 May 2022) and GraphPad (GraphPad, San Diego, CA, USA; available at: https://www.graphpad.com/quickcalcs/randomize1.cfm accessed on 1 May 2022) programs were used to create these figures.

## 3. Results

This study determined anaerobic, physical, and cognitive performance levels by intake of PLA, CAF, TAU and CAF*TAU supplements. In line with the data obtained, the effects of CAF, TAU and CAF*TAU on anaerobic, physical and cognitive performance were examined.

Supplementation of different conditions between repeated measures one-way ANOVA, as a result of PP (W) (F = _1.092_, *p* < 0.001, η_p_^2^ = 0.720), PP (W/kg) (F = _1.515_, *p* < 0.001, η_p_^2^ = 0.521), AP (W) (F = _2.512_, *p* < 0.001, η_p_^2^ = 0.548), MP (W) (F = _1.873_, *p* < 0.001, η_p_^2^ = 0.641), T_PP (s) (F = _2.412_, *p* < 0.001, η_p_^2^ = 0.672) and RPE (F = _1.424_, *p* = 0.048, η_p_^2^ = 0.345) revealed a statistically significant difference in the variables (Table 2).

One-way ANOVA test was applied to evaluate repeated measurements of different supplementation protocols. As a result, it was found that there was a statistically significant difference in the following variables: static dominant (F = _2.722_, *p* = 0.030, η_p_^2^ = 0.489), static nondominant (F = _1.935_, *p* = 0.047, η_p_^2^ = 0.542), dynamic dominant (F = _2.423_, v = 0.028, η_p_^2^ = 0.575), dynamic nondominant (F = _2.145_, *p* = 0.023, η_p_^2^ = 0.622), dynamic both leg (F = _3.834_, *p* = 0.015, η_p_^2^ = 0.660) and Illinois agility tests (F = _1.245_, *p* < 0.001, η_p_^2^ = 0.841) (Table 3).

One-way ANOVA test was applied to evaluate repeated measurements of different supplementation protocols. As a result, NR (ms) (F = _2.497_, *p* = 0.004, η_p_^2^ = 0.540), ICR (ms) (F = _5.210_, *p* < 0.001, η_p_^2^ = 0.610) and ICR (error rate) (F = _2.124_, *p* < 0.001, η_p_^2^ = 0.465) revealed a statistically significant difference in the variables (Figure 2 and Figure 3).

## 4. Discussion

This is the first study to examine the effects of single and combined CAF*TAU intake on Wingate performance in elite male boxers and to evaluate balance, agility and cognitive performance after fatigue. Our main findings were that co-ingestion of CAF*TAU on Wingate performance improved PP, AP, MP, T_PP and RPE performance compared to PLA. At the same time, a significant difference was observed in PP (W/kg) and AP parameters compared to CAF*TAU co-ingestion and intake of CAF and TAU separately. In addition, there was a significant difference in TAU intake, PP (W/kg), AP, and MP values according to CAF intake, CAF*TAU co-ingestion, PP, AP and MP values according to single dose of TAU and PP (W/kg), and AP values were significantly different. Compared to PLA, single or combined doses of CAF and TAU did not affect lactate. It was observed that the co-ingestion CAF*TAU was significantly different from the PLA values in the balance and agility performances evaluated after the Wingate test was applied. Moreover, both single and combined doses of CAF and TAU significantly improved agility performance compared to PLA. On the other hand, in the Stroop test applied in the evaluation of cognitive performance, the co-ingestion of CAF*TAU during the NR_(ms)_ task significantly improved the reaction time compared to TAU, CAF and PLA values. Again, it has been determined that the combined intake of CAF*TAU during the ICR_(ms)_ and ICR_(error rate)_ missions, which is considered difficult, significantly improved the reaction time and accuracy rate according to PLA values. In this study, a single dose of TAU improved PP, AP, MP and T_PP values in Wingate performance compared to PLA. However, TAU was shown to have ergogenic effects in this study, and mixed and inconsistent results regarding the improvement of anaerobic exercise performance of TAU intake have been reported in the literature [48,49].

It was very important in the preference of boxers as a population because the anaerobic system is more dominant in the energy systems used in boxers. To protect the decline of physical parameters after fatigue is created by anaerobic processes, and most importantly, reasoning, planning, quick decision making, etc. are very important for success in all sports branches and especially in boxing. As a result, the use of nutritional supplements that directly affect these processes constitutes the original nature of the study. In the current study, 3 g of a single dose of TAU, improved Wingate anaerobic power in elite male boxers. In the literature, it is stated that TAU improves aerobic exercise [50,51,52], but intake of TAU may be less effective during high-intensity exercise [21,53], as in the Wingate anaerobic power test. The possible reason of this situation can be explained as follows: via regulating lipid metabolism and activating the genes and proteins involved in mitochondrial biogenesis and respiratory function, a single dose of TAU enhances aerobic capacity [54,55,56]. While there is limited research on the effect of taurine on anaerobic and aerobic performance, more data are available on its effect on aerobics. Although little is known about how TAU affects anaerobic performance, previous research has shown that taurine can improve anaerobic performance [53,57]. However, the reason for this improvement has not been revealed. One of the main focuses of this study was to determine whether a single dose of TAU acutely affects anaerobic exercise performance. In support of our current findings, 3.75 g of TAU was found to increase the critical strength and tolerance of high-intensity exercise performed in recreationally active male participants [58]. In contrast, ingesting 1 g of TAU had no effect on improving Wingate anaerobic power in female athletes. After strenuous exercise, type II muscle fibers have exhausted 25% of the TAU concentration [59]. Greater benefits of a single dose of TAU are more likely to be seen in exercise practices involving repeated sprint experiments in the Wingate test. Warnock, Jeffries, Patterson and Waldron [22] reported that a single dose of TAU of approximately 4.3 g improved three-repetition mean peak power significantly. Collectively, these findings suggest that a single dose of 1 g TAU is ineffective to enhance a single Wingate anaerobic power performance. However, this dose increases aerobic endurance performance effectively [50]. Therefore, the use of doses indicates the need to be appropriately designed to improve exercise performance. For this reason, the importance of taking the dose at the right time, in the right amount and in the right form is emphasized to see the ergogenic effect.

While there is no study in the literature that previously evaluated agility and balance performance in the single dose of TAU intake, its evaluation after physical fatigue reveals a different unique aspect of our study. During agility and cognitive performance after the Wingate test protocol, while TAU significantly improved agility and cognitive performance (ICR_(ms)_ and ICR_(error rate)_ tasks) compared to PLA, there was no significant difference in balance performance. It is thought that the increased agility performance (10.85%) with a single dose of TAU is due to the increase in the activity of skeletal muscle glycolytic and oxidative enzymes (creatine kinase, lactate dehydrogenase and phosphofructokinase) that catalyze the energy required for muscle contraction [60]. In addition, it has been reported that the synthesis of cyclic AMP (cAMP), a stimulator of glycolytic enzyme (phosphorylase) activity, facilitates catecholamine secretion that occurs as a function of exercise intensity and during exercise [61]. The increase in cAMP synthesis is important for athletic performance because acute cAMP signaling in skeletal muscles also triggers muscle hypertrophy [62]. Finally, cAMP production can be directly stimulated by TAU via adenylate cyclase activation [63]. Taken together, these data suggest that the synergistic effect of TAU and cAMP during skeletal muscle contraction contributes to increased overall glycolytic enzyme activity through a possible release of catecholamines [60], thereby improving agility performance.

As a result of our literature review, it was understood that there was no study examining a single intake of TAU in the evaluation of cognitive performance. TAU is one of the most common ingredients in energy drinks, dietary supplements and decaffeinated energy drinks and is a nonessential sulfur-containing amino acid [39,64]. As the building blocks of proteins and precursors of neurotransmitters, amino acids are often added to energy drinks and supplements because increased availability of amino acids increases protein synthesis and neurotransmitter reserve [41,65,66], thereby affecting mood and cognitive performance. Therefore, cognitive tests were applied with combined purchases with more different ingredients (for example, Red Bull or energy drinks) and compared with PLA values. Despite the widespread inclusion of TAU in different products, few placebo-controlled studies have examined its effects on cognition. Studies advocating that TAU affects cognitive performance are available in the literature [67,68]. However, as we know, no study in the literature shows that TAU has no effect on cognitive performance. TAU’s possible improvement in cognitive performance can be explained as follows: TAU is found at high levels in the developing brain, as well as in the adult hippocampus [69], cerebellum [70], and hypothalamus [71]. TAU supports the proliferation of neural progenitor cells and synapse formation in brain regions required for long-term memory [69]. TAU stimulates action potentials in GABAergic neurons and is thought to improve cognitive performance by targeting the GABA_A_ receptor [72] and balancing the excitotoxic effects of glutamate [73] because the GABAergic system is crucial for decreasing the threshold for inputs on neurons in a homeostatic brain [74].

In this study, single ingestion of CAF improved PP, AP, MP and T_PP values in Wingate performance compared to PLA. As previously suggested [39,75], studies are showing that Wingate anaerobic power performance is enhanced by the single dose of CAF in both males [76] and females [26]. According to current findings, men and women both benefit equally from the ergogenic effects of CAF on Wingate performance [77]. In the current study, we found that the single intake of CAF improved Wingate anaerobic power in elite male boxers. During the agility and cognitive performance applied after the Wingate test protocol, while CAF significantly improved agility and cognitive performance (ICR_(ms)_ and ICR_(error rate)_ tasks) compared to PLA, there was no significant difference in balance performance. In addition to studies arguing that CAF affects cognitive performance [78,79,80], some studies suggest that CAF has no effect [81,82]. This study supported the potential ergogenic effect of caffeine. The possible reason for the potential ergogenic effect of caffeine can be explained as follows. It is believed that CAF exerts its effects on the CNS through the antagonism of adenosine receptors and leads to increases in neurotransmitter release, motor unit firing rates and pain suppression [83,84,85]. In the brain, adenosine and dopamine connect, and this may be a way to explain how vitality, endurance and performance outcome elements of motivation—and higher-level cognitive functions—are engaged in motor control [86]. Another CNS-related characteristic includes a decrease in skeletal muscle soreness and sensation of strength, which affects motivational elements to maintain effort during exercise by reducing the impression of effort during exercise. CAF provides a higher dopamine concentration, particularly in brain areas associated with attention [87]. Caffeine enhances sustained attention and alertness, as well as reducing signs of exhaustion, according to this neurobiological action. CAF ergogenicity also has some direct contributing effects on muscle. The most likely way that CAFs can benefit muscle contraction is through calcium ion (Ca^2+^) mobilization, which facilitates force production by each motor unit [86,88,89]. Fatigue caused by the gradual decrease in Ca^2+^ release can be alleviated after caffeine intake [89,90]. Similarly, CAF partially increases sodium/potassium pump activity (Na^+^/K^+^), potentially increasing the excitation–contraction coupling required for muscle contraction [91]. Since not all CNS aspects have been studied in detail, it should be considered that a placebo effect may also be present [92]. Consequently, CAF has a positive impact on performance in the brain [93], but more research is needed to uncover the precise mechanisms by which the CNS effect is produced.

Compared to PLA, the CAF*TAU combination was the most effective supplementation strategy for Wingate to improve anaerobic strength, balance, agility and cognitive performance. Since TAU increases Ca^++^ release in the sarcoplasmic reticulum [48] and CAF increases the activity of Na^+^/K^+^-ATP_ase_ channels [94], coadministration of CAF*TAU may improve the sensitivity of myofilaments that generate force in skeletal muscle by strengthening the muscles [22,95]. In addition, in the presence of CAF, in vitro TAU can enhance the strength gain of fast-twitch muscle fibers [96]. As neither CAF nor TAU alone improves more than their combined intake, it is thought that utilizing their synergistic effects may be a good strategy from an ergogenic point of view.

## 5. Conclusions

A combined intake of 6 mg/kg CAF and 3 g TAU improved power output during Wingate performance and subsequently agility, balance and cognitive performance in elite male boxers. Although using these supplements separately was effective, it was not more effective than taking them in combination. Sports scientists and dietitians may recommend the using of CAF*TAU together in sports branches (kickboxing, karate, taekwondo) where anaerobic power, agility, balance and cognitive performance are at the forefront. The experimental procedures used in this study can also be recommended for other sports branches, such as boxing, where anaerobic energy systems are dominant and where physical and cognitive performance is an important factor in success.

## Figures and Tables

**Figure 1 nutrients-14-04399-f001:**
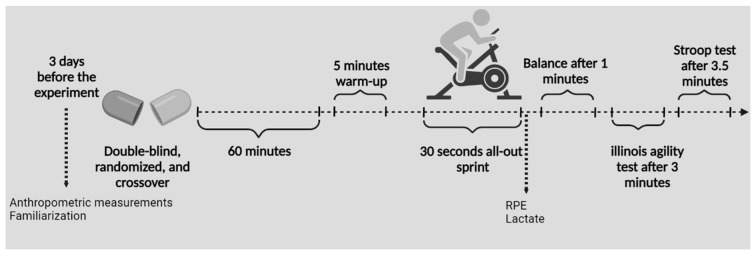
Experimental representation of the test protocol.

**Figure 2 nutrients-14-04399-f002:**
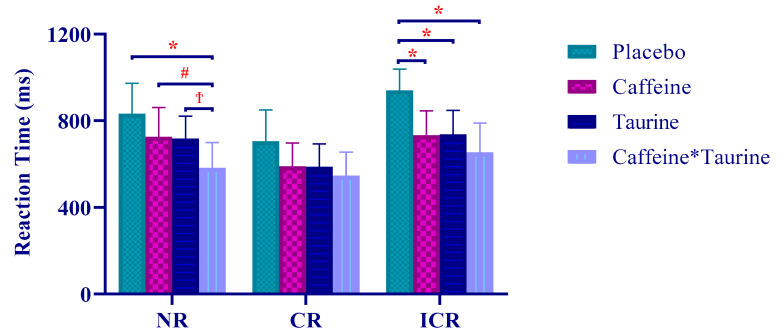
Stroop task reaction times in different supplement conditions. *: significantly different according to PLA values (*p* < 0.05), #: significantly different according to CAF values (*p* < 0.05), Ϯ: significantly different according to TAU values (*p* < 0.05).

**Figure 3 nutrients-14-04399-f003:**
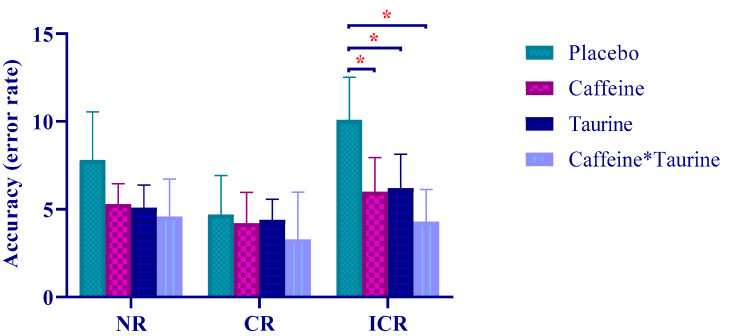
Stroop task accuracy (error rate) in different supplement conditions. *: significantly different according to PLA values (*p* < 0.05).

**Table 1 nutrients-14-04399-t001:** Characteristics of the participants.

Variables	Minimum	Maximum	Mean ± SD
Age (years)	18	24	22.14 ± 1.42
Sports age (years)	10	15	11.12 ± 1.12
Height (cm)	167	181	178 ± 6.45
Body mass (kg)	53	80	75.45 ± 8.95
BMI (kg/m^2^)	19	27	23.67 ± 3.12
Fat mass (%)	4.7	20.4	11.01 ± 5.12
Muscle mass (%)	79.6	95.3	89.17 ± 6.23

Abbreviations: BMI: body mass index; The minimum, maximum, mean, and standard deviation values of the participants’ characteristics are shown in table.

**Table 2 nutrients-14-04399-t002:** Physical and physiological responses were obtained from measurements under different conditions.

	PLA Mean ± SD	CAF Mean ± SD	TAU Mean ± SD	CAF*TAU Mean ± SD	F	*p*	η_p_^2^
PP (W)	633.02 ± 117.79	687.66 ± 125.38 *	653.64 ± 116.8 *	726.20 ± 144.43 *^Ϯ^	1.092	**<0.001**	0.720
PP (W/kg)	9.84 ± 1.09	10.65 ± 1.16 *	10.17 ± 1.08 *^#^	11.30 ± 1.42 *^#^^Ϯ^	1.515	**<0.001**	0.521
AP (W)	448.72 ± 82.89	469.52 ± 84.18 *	459.71 ± 87.74 *^#^	483.35 ± 87.20 *^#^^Ϯ^	2.512	**<0.001**	0.548
MP (W)	230.63 ± 34.01	275.18 ± 37.08 *	263.30 ± 40.55 *^#^	294.56 ± 49.25 *^Ϯ^	1.873	**<0.001**	0.641
T_PP (s)	13.90 ± 1.28	9.30 ± 1.20 *	9.60 ± 1.07 *	8.20 ± 1.03 *	2.412	**<0.001**	0.672
RPE	18.10 ± 1.42	17.20 ± 1.13	17.40 ± 0.84	16.60 ± 0.69 *	1.424	**0.048**	0.345
Lactate (mmol/L)	10.30 ± 1.16	9.30 ± 1.49	9.10 ± 1.09	8.30 ± 1.33	2.745	0.125	0.213

Abbreviations: PLA: placebo, CAF: caffeine, TAU: taurine, PP: peak power, AP: average power; MP: minimum power, T_PP: time to reach peak power, RPE: the rating of perceived exertion, mean ± SD: mean ± standard deviation, η_p_^2^: partial eta square coefficient, *: significantly different according to PLA values (*p* < 0.05), ^#^: significantly different according to CAF values (*p* < 0.05), ^Ϯ^: significantly different according to TAU values (*p* < 0.05) Bold texts indicate significant difference.

**Table 3 nutrients-14-04399-t003:** Balance and agility responses were obtained from measurements under different conditions.

	PLA Mean ± SD	CAF Mean ± SD	TAU Mean ± SD	CAF*TAU Mean ± SD	F	*p*	η_p_^2^
**Static Dominant**	836.70 ± 342.06	733.70 ± 350.50	711.20 ± 351.82	520.70 ± 256.39 *	2.722	**0.030**	0.489
**Static NonDominant**	825.10 ± 361.15	689.70 ± 314.01	701.50 ± 337.84	556.70 ± 220.95 *	1.935	**0.047**	0.542
**Static Both Leg**	663.50 ± 297.33	591.10 ± 278.98	603.10 ± 297.04	515.80 ± 225.55	4.968	0.489	0.123
**Dynamic Dominant**	1165.00 ± 208.61	1084.40 ± 258.24	1106.30 ± 169.47	910.00 ± 238.32 *	2.423	**0.028**	0.575
**Dynamic NonDominant**	1351.10 ± 396.55	1154.60 ± 350.95	1218.80 ± 394.61	918.00 ± 217.27 *	2.145	**0.023**	0.622
**Dynamic Both Leg**	1252.30 ± 275.44	1119.30 ± 256.32	1128.40 ± 227.62	1043.90 ± 245.62 *	3.834	**0.015**	0.660
**Illinois Agility Test**	18.79 ± 0.80	16.95 ± 0.73 *	16.75 ± 0.47 *	16.44 ± 0.52 *	1.245	**<0.001**	0.841

Abbreviations: PLA: placebo, CAF: caffeine, TAU: taurine, mean ± SD: mean ± standard deviation, η_p_^2^: partial eta square coefficient, *: significantly different according to PLA values (*p* < 0.05).

## Data Availability

Not applicable.

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
