# Peer review of "Does Single or Combined Caffeine and Taurine Supplementation Improve Athletic and Cognitive Performance without Affecting Fatigue Level in Elite Boxers? A Double-Blind, Placebo-Controlled Study"

_nutrients, 2022, doi:10.3390/nu14204399_

Round 1

Reviewer 1 Report

The manuscript "Does Single or Combined Caffeine and Taurine Supplementation Improve Athletic and Cognitive Performance Without Affecting Fatigue Level in Elite Boxers? : A Double-Blind, Placebo-Controlled Study" aims to evaluate the ergogenic effect of caffeine, taurine, and the co-ingestion of caffeine and taurine on the Wingate test in elite boxers and to determine the effects and possible changes in these parameters on balance, agility and cognitive performance after exercise. But there are several issues with the manuscript that I like to highlight:

1. In my opinion, the novelty of the manuscript is very modest and does not make a significant contribution to the field. This is especially true considering that many articles have already explored this or similar topic extensively.

2. The authors do not compare the differences between the subjects before the experiment was conducted and cannot ensure that the subjects are balanced and comparable and this is a major flaw.

3. The authors should detail the criteria for inclusion of study participants, for example, ten years of experience in boxing. What is the basis for selection and whether the length of years biased the study results.  

4. CAF and TAU should be given in full the first time appeared in the text.

5. Abbreviations should be listed below the table.

6. Figure 2 and 3 and Table 4 show the same content.

Author Response

Thank you very much for your valuable comments and outstanding scientific contribution to our study. We improved the manuscript according to your helpful comments. In addition, we would like to express our gratitude and thanks to the reviewers and editor who read the study meticulously and carefully, as authors, for their efforts and good wishes. All the corrections and additions throughout the manuscript are shown in yellow color throughout the manuscript.

Reviewer 2 Report

First, congratulations for the courage and stamina to undergo on a clinical trial, specially adding up evidence on sports supplements which always require means and methods to ensure that the study is worth. 

The study was well design therefore comments will be mostly focused in improving the manuscript to ensure that this can be further one reference work in future reviews about this subject. 

Abstract

I would suggest you start it with a 1 line background about the use of these dietary supplements in improving performance. 

Introduction

Update references for the first paragraph, consider works from Robert Maughan 

Line 44 caffeine above 6mg/kg can have side effects, normally the dosage is 3 to 6mg/kg 

Line 68 You say “recent studies” but you only reference one , you can say “recent results”

I would suggest you shorten the introduction, cutting some information on these two supplements (leaving for discussion text) and describe better the features of boxing sports that can be improved by taurine and caffeine. 

Methods 

Line 97 Sample characteristics can be in results , you can say in 2.1. how were the participants recruited and as presented the inclusion and exclusion criteria

Also, in exclusion, did you refer the possibility that any participant did not had any supplement before? If not, consider and refer it as a limitation due to the possible placebo effect risk from “taking something”

Line 131 “take supplements …” this sentence is a little weirdly placed here 

Line 137 you can gave the participants a meal plan. But did you studied their food intake before and after the study to ensure that it was not different? 

Results 

Line 228 So table 1 would be sample characteristics, I did not find it in the text. It is like a caption below the table 

I would suggest you describe clearly the comparison between supplements and placebo in the main tests in this section because in tables is not so clear. 

Lines 290-294 this is a little confusing, you go further and then back to anaerobic

Line 375 It has been previously suggested in a drawback study with caffeine in cyclists, you can refer it 

Why cognitive performance was tested in your study? Support it because it is important to explain the tests used in addition to the Wingate protocol. 

Conclusion

You can address further applications of this combination specifically in combat sports or similar. 

The use of this brand is random? Or the supplements and gelatin caps were supplied by Myprotein? Considering that you refer the brand, please clarify conflicts of interest. 

Author Response

(The authors gave the same response as above.)

Round 2

Reviewer 1 Report

Although the authors made the study subjects as homogeneous as possible through training, diet, etc., the authors did not compare whether there was a difference in performance between the groups before the study began. If this difference could not be controlled for, the difference in the results of the experiment could not be said to be due to the intervention factors.

Author Response

First of all, as we mentioned in the first revision, all external factors were tried to be kept constant in order to reduce the bias of the experimental procedure. A homogeneous group was formed by considering all conditions such as training age, nutrition, living conditions, accommodation conditions, training programs. One group was measured at different times in four conditions (supplement: placebo, taurine, caffeine, caffeine*taurine) to see whether combined or single intake of caffeine and taurine affected their physical and cognitive processes. In order not to affect the results to be obtained in the measurements, measurements were made on the same groups at different times, not on different groups. Indeed, if we had done it in different groups, as you said, the results might have been affected due to the performance differences among the participants. However, in order to avoid this bias, we determined a single group and minimized the bias of the measurements. While doing this study, taking into account the circadian rhythm, sleep, light, food intake, ambient temperature, etc. zeitgebers were checked. Measurements were made at the same time of the day and great attention was paid to supplement intake. In all experimental procedures, the experimental flow is the same, and it has been tried to be the same by the experts. In conclusion, we understood your concerns very well and tried to answer them with this revision. I hope we were successful.

Reviewer 2 Report

Congratulations for the extensive revision you have done, they improved significantly your work. I have some extra comments related to references and claims: 

So, on the first paragraph, line 40-43, I would suggest a better reference considering that you are not referring to caffeine rather to supplement usage. Therefore you have a good consensus document from IOC http://dx.doi.org/10.1136/bjsports-2018-099027

Line 44, I am not sure you can say the evidence on caffeine effect on strenght is inconsistent considering the meta-analysis you have. You could rather say it is relevant to increase evidence on several sports, including combat sports because physical effort can be very diverse within sports. 

Lines 415, considering that each sport is different maybe it is too ambitious and not your main conclusion. 

Author Response

  1. Congratulations for the extensive revision you have done, they improved significantly your work. I have some extra comments related to references and claims:

Response: Thank you for your positive contribution to our work. With your valuable suggestions, our work has become a better manuscript. We are grateful.

  1. So, on the first paragraph, line 40-43, I would suggest a better reference considering that you are not referring to caffeine rather to supplement usage. Therefore you have a good consensus document from IOC http://dx.doi.org/10.1136/bjsports-2018-099027.

Response: It has been revised and corrected. Reference is used where relevant. We couldn't reach the full study in the first revision. It was very good that you sent a DOI number.

  1. Line 44, I am not sure you can say the evidence on caffeine effect on strength is inconsistent considering the meta-analysis you have. You could rather say it is relevant to increase evidence on several sports, including combat sports because physical effort can be very diverse within sports.

Response:  It was exactly as you expressed it when we wanted to write it. The sentence has been revised and changed. The phrase expressed was "The findings of studies examining caffeine's ability to increase maximum strength and repetition against fatigue are inconsistent." It has been modified as follows: "Findings of studies examining the ability of caffeine to increase maximum strength and repetition against fatigue are considered appropriate to increase the evidence for various sports, including combat sports".

  1. Lines 415, considering that each sport is different maybe it is too ambitious and not your main conclusion.

Response: Indeed, our result is given by generalization. In the light of our main findings, our conclusion was given differently. The following statement is "The experimental procedures used in this study can also be recommended for other sports." has been changed. "The experimental procedures used in this study can also be recommended for other sports, such as boxing, where anaerobic energy systems are dominant and physical and cognitive performance is an important factor in success." expressed in this sentence.
